# FULLY DECENTRALIZED MODEL-BASED POLICY OPTIMIZATION WITH NETWORKED AGENTS

## ABSTRACT

Model-based RL is an effective approach for reducing sample complexity. However, when it comes to multi-agent setting where the number of agent is large, the model estimation can be problematic due to the exponential increased interactions. In this paper, we propose a decentralized model-based reinforcement learning algorithm for networked multi-agent systems, where agents are cooperative and communicate locally with their neighbors. We analyze our algorithm theoretically and derive an upper bound of performance discrepancy caused by model usage, and provide a sufficient condition of monotonic policy improvement. In our experiments, we compare our algorithm against other strong multi-agent baselines and demonstrate that our algorithm not only matches the asymptotic performance of model-free methods but also largely increases its sample efficiency.

## 1 INTRODUCTION

Many real world problems, such as autonomous driving, wireless communications, multi-player games can be modeled as multi-agent RL problems, where multiple autonomous agents coexist in a common environment, aiming to maximize its individual or team reward in the long term by interacting with the environment and other agents. Unlike single-agent tasks, multi-agent tasks are more challenging, due to partial observations and unstable environments when agents update their policies simultaneously. Therefore, there are hardly any one-fits-all solutions for MARL problems. Examples include networked systems control (NSC) (Chu et al., 2020), in which agents are connected via a stationary network. They perform decentralized control based on its local observations and messages from connected neighbors. Examples of networked systems include connected vehicle control (Jin & Orosz, 2014), traffic signal control (Chu et al., 2020), etc.

Despite the success of multi-agent reinforcement (RL) algorithms, their performance relies on a massive amount of model usage. Typically, a multi-agent RL algorithm needs millions of interaction with the environment to converge. On the other hand, model-based reinforcement learning (MBRL) algorithms, which utilize predictive models of the environment to help data collection, are empirically more data-efficient than model-free approaches. Although model inaccuracy performs as a bottleneck of policy quality in model-based algorithms, we can still learn a good policy with an imperfect model (Luo et al., 2019), especially combined with the trick of branched rollout (Janner et al., 2019) to limit model usage. Experimentally, MuZero (Schrittwieser et al., 2020), a model-based RL algorithm, succeeded in matching the performance of AlphaZero on Go, chess and shogi, and becomes state-of-the-art on Atari games. Model-based MARL is not fully investigated. Existing MB-MARL algorithms either limit their field of research on specific scenario, e.g. two-player zero-sum Markov game (Zhang et al., 2020) or pursuit evasion game (Bouzy & Métivier, 2007), or use tabular RL method (Bargiacchi et al., 2021). MB-MARL for multi-agent MDPs is still an open problem to be solved (Zhang et al., 2019), with profound challenges such as scalability issues caused by large state-action space and incomplete information of other agents' state or actions.

In this paper, we develop decentralized model-based algorithms on networked systems, where agents are cooperative, and able to communicate with each other. We use localized models to predict future states, and use communication to broadcast their predictions. To address the issue of model error, we adopt branched rollout (Janner et al., 2019) to limit the rollout length of model trajectories. In the policy optimization part, we use decentralizd PPO (Schulman et al., 2017) with a extended value function. At last, we analyze these algorithms theoretically to bound the performance discrepancy

between our method and its model-free, centralized counterpart. At last, we run these algorithms in traffic control environments (Chu et al., 2020; Vinitsky et al., 2018) to test the performance of our algorithm. We show that our algorithm increases sample efficiency, and matches the asymptotic performance of model-free methods.

In summary, our contributions are three-fold. Firstly, we propose an algorithmic framework, which is a fully decentralized model-based reinforcement learning algorithm, which is named as **D**ecentralized **M**odel-based **P**olicy **O**ptimization (DMPO). Secondly, we analyze the theoretical performance of our algorithm. Lastly, empirical results on traffic control environments demonstrate the effectiveness of DMPO in reducing sample complexities and achieving similar asymptotic performance of model-free methods.

## 2   RELATED WORK

Model-based methods are known for their data efficiency (Kaelbling et al., 1996), especially compared with model-free algorithms. There is a vast literature on the theoretical analysis of model-based reinforcement learning. In a single-agent scenario, monotonic improvement of policy optimization has been achieved (Luo et al., 2019; Sun et al., 2018), and a later work improved the performance of model-based algorithms by limiting model usage (Janner et al., 2019). But these analysis is restricted to single-agent scenarios, whereas ours addresses multi-agent problems.

On the other hand, Networked System Control (NSC) (Chu et al., 2020) is a challenging setting for MARL algorithm to take effect. Some multi-agent algorithms falls into centralized training decentralized execution (CTDE) framework. For example, QMIX (Rashid et al., 2018) and COMA (Foerster et al., 2018) all use a centralized critic. In a large network, however, centralized training might not scale. In many scenarios, only fully decentralized algorithms can be used. Zhang et al. (2018) proposed an algorithm of NSC that can be proven to converge under linear approximation. Qu et al. (2020a) proposed truncated policy gradient, to optimize local policies with limited communication. Baking in the idea of truncated $Q$-learning in (Qu et al., 2020a), we generalize their algorithm to deep RL, rather than tabular RL. Factoring environmental transition into marginal transitions can be seen as factored MDP. Guestrin et al. (2001) used Dynamic Bayesian Network to predict system transition. Simao & Spaan (2019) proposed a tabular RL algorithm to ensure policy improvement at each step. However, our algorithm is a deep RL algorithm, enabling better performance in general tasks.

There are some works on applying model-based methods in MARL settings. A line of research focuses on model-based RL for two-player games. For example, Brafman & Tennenholtz (2000) solved single-controller-stochastic games, which is a certain type of two-player zero-sum game; Bouzy & Métivier (2007) performed MB-MARL in the pursuit evasion game; Zhang et al. (2020) proved that model-based method can be nearly optimally sample efficient in two-player zero-sum Markov games. Bargiacchi et al. (2021) extended the concept of prioritized sweeping into a MARL scenario. However, this is a tabular reinforcement algorithm, thus unable to deal with cases where state and action spaces are relatively large, or even continuous. In contrast to existing works, our algorithm is not only applicable to more general multi-agent problems, but is also the first fully decentralized model-based reinforcement learning algorithm.

## 3   PROBLEM SETUP

In this section, we introduce multi-agent networked MDP and model-based networked system control.

**Networked MDP**   We consider environments with a graph structure. Specifically, $n$ agents coexist in an underlying undirected and stationary graph $\mathcal{G} = (\mathcal{V}, \mathcal{E})$. Agents are represented as a node in the graph, therefore $\mathcal{V} = \{1, ..., n\}$ is the set of agents. $\mathcal{E} \subset \mathcal{V} \times \mathcal{V}$ comprises the edges that represent the connectivity of agents. Agents are able to communicate along the edges with their neighbors. Let $N_i$ denote every neighbor of agent i, and $\bar{N}_i = N_i \cup \{i\}$. Furthermore, let $N_i^\kappa$ denote the $\kappa$-hop neighborhood of $i$, i.e. the nodes whose graph distance to $i$ is less than or equal to $\kappa$. For the simplicity of notation, we also define $N_{-i}^\kappa = \mathcal{V} \setminus N_i^\kappa$.

The corresponding networked MDP is defined as $(\mathcal{G}, \{\mathcal{S}_i, \mathcal{A}_i\}_{i\in\mathcal{V}}, p, r)$. Each agent $i$ have their local state $s_i \in \mathcal{S}_i$, and perform action $a_i \in \mathcal{A}_i$. The global state is the concatenation of all local states: $s = (s_1, ..., s_n) \in \mathcal{S} := \mathcal{S}_1 \times ... \times \mathcal{S}_n$. Similarly, the global action is $a = (a_1, ..., a_n) \in \mathcal{A} := \mathcal{A}_1 \times ... \times \mathcal{A}_n$. For the simplicity of notation, we define $s_{N_i}$ to be the local states of every agent in $N_i$, that is, given $N_i = \{j_1, ..., j_c\}$, then $s_{N_i} = (s_{j_1}, ..., s_{j_c})$. $a_{N_i}, s_{N_i^\kappa}, a_{N_i^\kappa}$ are defined similarly. The transition function is defined as: $p(s'|s, a) : \mathcal{S} \times \mathcal{A} \to \mathcal{S}$. Each agent possess a localized policy $\pi_i^{\theta_i}(a_i|s_{\bar{N}_i})$ that is parameterized by $\theta_i \in \Theta_i$, meaning the local policy is dependent only on states of its neighbors and itself. We use $\theta = (\theta_1, ..., \theta_n)$ to denote the tuple of localized policy parameters, and $\pi^\theta(a|s) = \prod_{i=1}^n \pi_i^{\theta_i}(a_i|s_{\bar{N}_i})$ denote the joint policy. We also assume that reward functions is only dependent on local state and action: $r_i(s_i, a_i)$, and the global reward function is defined to be the average reward $r(s, a) = \frac{1}{n}\sum_{i=1}^n r_i(s_i, a_i)$.

The goal of reinforcement learning is to maximize the expected sum of discounted rewards, denoted by $\eta$:

$$\pi^{\theta^*} = \arg\max_{\pi^\theta} \eta[\pi^\theta] = \arg\max_{\pi^\theta} \mathbb{E}_{\pi^\theta}\Big[\sum_{t=0}^\infty \gamma^t \cdot \frac{1}{n}\sum_{i=1}^n r_i(s_t, a_t)\Big], \quad (1)$$

where $\gamma \in (0, 1)$ is the temporal discount factor. We define the stationary distribution under policy $\pi$ to be $d_\pi(s)$.

**Independent Networked System** Networked system may have some extent of locality, meaning in some cases, local states and actions do not affect the states of distant agents. In such systems, environmental transitions can be factorized, and agents are able to maintain local models to predict future local states. We define Independent Networked System (INS) as follows:

**Definition 1.** *An environment is an Independent Networked System (INS) if:*

$$p(s'|s, a) = \prod_{i=1}^n p_i(s_i'|s_{\bar{N}_i}, a_i), \forall s', s \in \mathcal{S}, a \in \mathcal{A}.$$

INS might be an assumption that is too strong to hold. However, for the dynamics that cannot be factorized, we can still use an INS to approximate it. Let $D_{TV}$ denote the total variation distance between distributions, we have the following definition:

**Definition 2.** *($\xi$-dependent) Assume there exists an Independent Networked System $\bar{p}$ such that $\bar{p}(s'|s, a) = \prod_{i=1}^n p_i(s_i'|s_{\bar{N}_i}, a_i)$. An environment is $\xi$-dependent, if:*

$$\sup_{s,a} D_{TV}\Big(p(s'|s, a)\|\bar{p}(s'|s, a)\Big) = \sup_{s,a}\frac{1}{2}\sum_{s'\in\mathcal{S}}|p(s'|s, a) - \bar{p}(s'|s, a)| \leq \xi.$$

To explain the intuition behind this definition, we point out that $\xi$ is actually the lower bound of model error when we use local models $\hat{p}(s_{\bar{N}_i}, a_i)$. Recall that $p(s'|s, a)$ is the real environment transition, $\bar{p} = \prod_{i=1}^n p_i(s_i'|s_{\bar{N}_i}, a_i)$ is the product of marginal environment transitions, and $\hat{p}(s, a) = \prod_{i=1}^n \hat{p}_i(s_i'|s_{\bar{N}_i}, a_i)$ is the product of model transitions. Then the universal model error $D(p\|\hat{p})$ can be divided into two parts: dependency bias $D(p\|\bar{p})$ and model error $D(\bar{p}\|\hat{p})$:

$$D(p\|\hat{p}) \leq D(p\|\bar{p}) + D(\bar{p}\|\hat{p}).$$

Then for a $\xi$-dependent system, when models become very accurate, meaning $D(\bar{p}\|\hat{p}) \approx 0$, $\sup D(p\|\hat{p}) \approx \sup D(p\|\bar{p}) = \xi$. While $D$ can be any appropriate distance metric, we use the TV-distance hereafter for the ease of presentation. In the following, we develop theory under both INS and $\xi$-dependent scenarios.

## 4 DECENTRALIZED MODEL-BASED POLICY OPTIMIZATION

In this section, we formally present Decentralized Model-based Policy Optimization (DMPO), which is a fully decentralized model-based reinforcement learning algorithm. Compared with independent multi-agent PPO, DMPO is augmented in three ways: localized model, policy with one-step communication, and extended value function. We introduce the detail of localized model in 4.1. Policy and value functions are introduced in 4.2. The illustration of our algorithm is given in Figure 1. All the components mentioned above are analyzed in Section 5. We argue that under certain conditions, our algorithm ensures monotonic policy improvement.

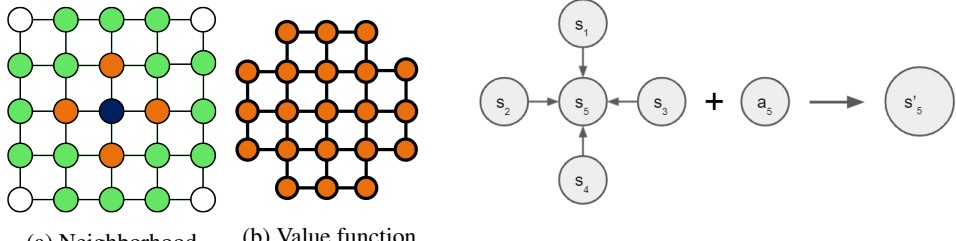

(a) Neighborhood  (b) Value function

(c) Graph convolutional model

Figure 1: (a) presents the concept of neighborhood. If agent $i$ is the node in purple, then purple and orange is $\bar{N}_i$, and combination of purple, orange and green is $N_i^3$. (b) explains that extended value function takes $s_{N_i^\kappa}$ as input, here $\kappa = 3$. (c) is the illustration of graph convolutional model.

## 4.1 DECENTRALIZED PREDICTIVE MODEL

To perform decentralized model-based learning, we let each agent maintain a localized model. The localized model can observe the state of 1-hop neighbor and the action of itself, and the goal of a localized model is to predict the information of the next timestep, including state, reward and done. This process is denoted by $\hat{p}_i(s'_i, r'_i, d'_i | s_{\bar{N}_i}, a_i)$.

We implement a localized model with graph convolutional networks (GCN). Recall that agents are situated in a graph $\mathcal{G} = (\mathcal{V}, \mathcal{E})$. In the first step, a node-level encoder encodes local state into node embedding,

$$h_i^0 = f_i^{encode}(s_i). \tag{2}$$

Then we perform one step of graph convolution as follows,

$$h_{(i,j)} = f_{(i,j)}^{edge}(h_i^0, h_j^0), \quad h_i^1 = f_i^{node}\left(\sum_{e=(i,j)} h_{(i,j)}, a_i\right). \tag{3}$$

In this way, $h_i^1$ is dependent only on $s_{\bar{N}_i}$ and $a_i$. Finally, a node-level decoder generates the prediction of state, reward and done from $h_i^1$ as follows:

$$s'_i = f_i^{state}(h_i^1) + s_i, \quad r'_i = f_i^{reward}(h_i^1), d'_i = f_i^{done}(h_i^1). \tag{4}$$

Note that we predict the next state with a skip connection, because empirically, it's more efficient to predict the change of the state rather than the state itself.

In practice, the data are all stored locally by each agent. Data that are collected in the environment by agent $i$ is denoted as $\mathcal{D}_i^{env}$, and those generated by predictive model is denoted as $\mathcal{D}_i^{model}$.

Scaling model-based methods into real tasks can result in decreased performance, even if the model is relatively accurate. One main reason is the compound modeling error when long model rollouts are used, and model error compound along the rollout trajectory, making the trajectory ultimately inaccurate. To reduce the negative effect of model error, we adopt a branched rollout scheme proposed in (Janner et al., 2019). In branched rollout, model rollout starts not from an initial state, but from a state that was randomly selected from the most recent environmental trajectory $\tau$. Additionally, the model rollout length is fixed to be $T$. This scheme is shown to be effective in reducing the negative influence of model error both theoretically and empirically.

To deal with the bias of model trajectories, at each model rollout, we allow the algorithm to fall back to the real trajectory with probability $1 - q_0$, where $q_0$ is a hyperparameter. We describe the detailed framework of model usage and experiment storage in Algorithm 1.

## 4.2 PROXIMAL POLICY OPTIMIZATION WITH EXTENDED VALUE FUNCTION

To optimize the policies, we need to adopt an algorithm that can exploit network structure, whilst remaining decentralized. Independent RL algorithms that observes only local state are fully decentralized, but they often fail to learn an optimal policy. Centralized algorithms that utilize centralized

---

**Algorithm 1:** Decentralized Model-based Policy Optimization (DMPO) for MARL

---

**Input:** hyperparameters: rollout length $T$, truncation radius $\kappa$

1: Initialize the model $p_i^{\psi_i}$, actor $\pi_i^{\theta_i}$ and critic $V_i^{\phi_i}$.
2: Initialize replay buffers $\mathcal{D}_i^{env}$ and $\mathcal{D}_i^{model}$.
3: **for** $M$ iterations **do**
4:     Perform environmental rollout together, and each agent $i$ collect trajectory information $\tau_i$.
5:     **for** $i$ in $N$ agents **do**
6:         $D_i^{env} = D_i^{env} \cup \{\tau_i\}$.
7:         Train $p_i^{\psi_i}$ on $D_i^{env}$.
8:     $D_i^{model} = \emptyset$.
9:     **for** $B$ inner iterations: **do**
10:         Generate a random number $q \sim U(0, 1)$.
11:         **if** $q > q_0$ **then**
12:             $\mathcal{D}_i^{model} = \tau_i$. {Fall back to real trajectory with probability $1 - q_0$.}
13:         **else**
14:             **for** $R$ rollouts, $s \in \tau$ **do**
15:                 Perform $T$-step model rollout starting from $s$ using policy $p_{\psi_*}$, append to $D_*^{model}$.
16:         **for** $G$ steps, $i = 0, ..., n - 1$ **do**
17:             Take a step along the gradient to update $\pi_i^{\theta_i}$ and critic $V_i^{\phi_i}$ on $D_*^{model}$

---

critics often achieve better performance than decentralized algorithms, but they might not scale to large environments where communication costs are expensive.

We propose Proximal Policy Optimization with *extended value function*, which is defined as $V_i(s_{N_i^\kappa}) = \mathbb{E}_{s_{N_{-i}^\kappa} \sim d_\pi}[\sum_{t=0}^\infty r_i^t | s_{N_i^\kappa}^0 = s_{N_i^\kappa}], i \in \mathcal{V}$. The intuition behind extended value function comes from (Qu et al., 2020a), where *truncated Q-function* $Q(s_{N_i^\kappa}, a_{N_i^\kappa})$ is initially proposed. In 5.3, we prove that $V_i(s_{N_i^\kappa})$ is a good approximation of $V_i(s)$, with a difference decreasing exponentially with $\kappa$.

To generate the objective for extended value function, or return $R_i$, we use reward-to-go technique. However, because model rollout is short, standard reward-to-go returns would get a biased estimation of $V_i$. To resolve this issue, we add the value estimation of the last state to the return. In this way, with a local trajectory $\tau_i = \{(s_i^t, a_i^t, r_i^t, (s')_i^t, d_i^t, \log \pi_i^t), t = 0, 1, ..., T - 1\}$, the objective of $V_i^t(s_{N_i^\kappa})$ is

$$R_i^t = \sum_{l=0}^{T-t-1} \gamma^l r_i^{t+l} + V_i^{\phi_i}\left[(s')_{N_i^\kappa}^{T-1}\right], \tag{5}$$

and the loss of value function is defined as $\mathcal{L}_i^{value} = \frac{1}{m} \sum_{m \in \mathcal{D}_i^{model}} \left[V_i^{\phi_i}(s_{N_i^\kappa}^m) - R_i^m\right]^2$. In policy training, extended value functions $V_i$ are reduced via communication to their $\kappa$-hop neighbors to generate an estimation of global value function,

$$\tilde{V}_i^t = \frac{1}{n} \sum_{j \in N_i^\kappa} \tilde{V}_j^t, \tag{6}$$

and advantages $\hat{A}_i$ are computed on $\tilde{V}_i$ with generalized advantage estimation (GAE) (Schulman et al., 2015) for policy gradient update. The surrogate loss function of a DMPO agent is defined as

$$\mathcal{L}_i^{policy} = \frac{1}{m} \sum_{m \in \mathcal{D}_i^{model}} \min\left(\frac{\pi_i^{\theta_i}(a_i^t|s_{\bar{N}_i}^t)}{\pi_i^{\theta_i^k}(a_i^t|s_{\bar{N}_i}^t)} \hat{A}_i(s_{V_i^\kappa}), g(\epsilon, \hat{A}_i(s_{V_i^\kappa}))\right), \tag{7}$$

similar to PPO-Clip loss.

The communication of $\kappa$ step might seem costly, yet information of $N_i^\kappa$ is only used in the training phase. We argue that in the training phase, algorithms are less sensitive with latency than execution. Furthermore, since model-based learning can effectively increase sample efficiency, we might tolerate more communication.

## 5 THEORETICAL ANALYSIS

In this section, we analyze DMPO theoretically. In 5.2, we derive a bound between the true returns and the returns under a model $\hat{p}$ in a networked system. In 5.3, we prove that extended value function $V_i(s_{N_i^\kappa})$ is a good approximation of $V_i(s)$, and with extended value function, the true policy gradient can also be approximated.

### 5.1 BACKGROUND: MONOTONIC MODEL-BASED POLICY OPTIMIZATION

Let $\eta[\pi]$ denote the returns of the policy in the true environment, $\hat{\eta}[\pi]$ denote the returns of the policy under the approximated model. To analyze the difference between $\eta[\pi]$ and $\hat{\eta}[\pi]$, we need to construct a bound

$$\eta^p[\pi] \geq \hat{\eta}^{\hat{p}}[\pi] - C(p, \hat{p}, \pi, \pi_D), \tag{8}$$

where $C$ is a non-negative function, and $\pi_D$ is the data-collecting policy. According to equation 8, if every policy update ensures an improvement of $\hat{\eta}[\pi]$ by at least $C$, $\eta[\pi]$ will improve monotonically. This inequality was first presented in single agent domain (Janner et al., 2019). In this work, we extend this to the multi-agent networked system, aiming to achieve monotonic team reward improvement.

In this work, we let $\pi$ indicate a collective policy $\pi = [\pi_1, ..., \pi_n]$, and the model $\hat{p}$ be an INS $\hat{p}(s'|s,a) = \prod_{i=1}^n \hat{p}_i(s'_i|s_{\bar{N}_i}, a_i)$ that approximating the true MDP. In DMPO, each agent learns a localized model $\hat{\pi}_i$, policy $\pi_i(|s_{N_k})$, critic $V_i(s_{N_i^\kappa})$, making it never a trivial extension. We give the detailed analysis in 5.2.

### 5.2 ANALYSIS OF RETURNS BOUND

In model-based learning, different rollout schemes can be chosen. The *vanilla rollout* assumes that models are used in an infinite horizon. The *branched rollout* performs a rollout from a state sampled by a state distribution of previous policy $\pi_D$, and runs $T$ steps in $\hat{\pi}$ according to $\pi$. Based on different rollout schemes, we can construct two lower bounds. Under *vanilla rollout*, real return and model return can be bounded by model error and policy divergence. Formal results are presented in Theorem 1. The detailed proof is deferred to Appendix C.

**Theorem 1.** *Consider an independent networked system. Denote local model errors as $\epsilon_{m_i} = \max_{s_{\bar{N}_i}, a_i} D_{TV}[p_i(s'_i|s_{\bar{N}_i}, a_i)\|\hat{p}_i(s'_i|s_{\bar{N}_i}, a_i)]$, and divergences between the data-collecting policy and evaluated policy as $\epsilon_{\pi_i} = \max_{s_{\bar{N}_i}} D_{TV}[\pi_D(a_i|s_{\bar{N}_i})\|\pi(a_i|s_{\bar{N}_i})]$. Assume the upper bound of rewards of all agents is $r_{\max}$. Let $\eta^p[\pi_1, ..., \pi_n]$ denote the real returns in the environment. Also, let $\eta^{\hat{p}}[\pi_1, ..., \pi_n]$ denote the returns estimated in the model trajectories, and the states and actions are collected with $\pi_D$. Then we have:*

$$|\eta^p[\pi_1, ..., \pi_n] - \eta^{\hat{p}}[\pi_1, ..., \pi_n]| \leq \frac{2r_{\max}}{1-\gamma} \sum_{i=1}^n \Big[\frac{\epsilon_{\pi_i}}{n} + (\epsilon_{m_i} + 2\epsilon_{\pi_i}) \cdot \sum_{k=0}^\infty \gamma^{k+1} \frac{|\bar{N}_i^k|}{n}\Big].$$

Intuitively, the term $\sum_{k=0}^\infty \gamma^{k+1} \frac{|\bar{N}_i^k|}{n}$ would be in the same magnitude as $\frac{1}{1-\gamma}$, which might be huge given the choice of $\gamma$, making the bound too loose to be effective. To make tighter the discrepancy bound in Theorem 1, we adopt the *branched rollout* scheme. The *branched rollout* enables a effective combination of model-based and model-free rollouts. For each rollout, we begin from a state sample from $d_{\pi_D}$, and run $T$ steps in each localized $\hat{\pi}_i$. When branched rollout is applied in an INS, Theorem 2 gives the returns bound.

**Theorem 2.** *Consider an independent networked system. Denote local model errors as $\epsilon_{m_i} = \max_{s_{\bar{N}_i}, a_i} D_{TV}[p_i(s'_i|s_{\bar{N}_i}, a_i)\|\hat{p}_i(s'_i|s_{\bar{N}_i}, a_i)]$, and divergences between the data-collecting policy and evaluated policy as $\epsilon_{\pi_i} = \max_{s_{\bar{N}_i}} D_{TV}[\pi_D(a_i|s_{\bar{N}_i})\|\pi(a_i|s_{\bar{N}_i})]$. Assume the upper bound of rewards of all agents is $r_{\max}$. Let $\eta^p[\pi_1, ..., \pi_n]$ denote the real returns in the environment. Also, let $\eta^{branch}[\pi_1, ..., \pi_n]$ denote the returns estimated via $T$-step branched rollout scheme. Then we have:*

$$|\eta^p[\pi_1, ..., \pi_n] - \eta^{branch}[\pi_1, ..., \pi_n]| \leq \frac{2r_{\max}}{1-\gamma} \sum_{i=1}^n \Big[\epsilon_{m_i} \cdot \Big(\sum_{k=0}^{T-1} \gamma^{k+1} \frac{|\bar{N}_i^k|}{n}\Big) + \epsilon_{\pi_i} \cdot \Big(\sum_{k=T}^\infty \gamma^{k+1} \frac{|\bar{N}_i^k|}{n}\Big)\Big]$$

Comparing the results in Theorem 1 and 2, we can see that branched rollout scheme reduced the co-efficient before $\epsilon_{m_i}$ from $\sum_{k=0}^{\infty} \gamma^{k+1} \frac{|\bar{N}_i^k|}{n} \leq \frac{\gamma}{1-\gamma}$ to $\sum_{k=0}^{T-1} \gamma^{k+1} \frac{|\bar{N}_i^k|}{n} \leq \sum_{k=0}^{T-1} \gamma^{k+1} = \frac{\gamma(1-\gamma^T)}{1-\gamma}$. This reduction explains that empirically, branched rollout brings better asymptotic performance. Also, if we set $T = 0$, this bound turn into a model-free bound. This indicates that when $\epsilon_{m_i}$ is lower than $\epsilon_{\pi_i}$ allowed by our algorithm, a model might increase the performance.

In reality, not every system satisfies the definition of INS. Yet we can generalize Theorem 2 into a $\xi$-dependent system.

**Corollary 1.** *Consider an $\xi$-dependent networked system. Denote local model errors as $\epsilon_{m_i} = \max_{s_{\bar{N}_i}, a_i} D_{TV}[p_i(s_i'|s_{\bar{N}_i}, a_i)\|\hat{p}_i(s_i'|s_{\bar{N}_i}, a_i)]$, and divergences between the data-collecting policy and evaluated policy as $\epsilon_{\pi_i} = \max_{s_{\bar{N}_i}} D_{TV}[\pi_D(a_i|s_{\bar{N}_i})\|\pi(a_i|s_{\bar{N}_i})]$. Assume the upper bound of rewards of all agents is $r_{\max}$. Let $\eta^p[\pi_1, ..., \pi_n]$ denote the real returns in the environment. Also, let $\eta^{branch}[\pi_1, ..., \pi_n]$ denote the returns estimated via $T$-step branched rollout scheme. Then we have:*

$$|\eta^p[\pi_1, ..., \pi_n] - \eta^{branch}[\pi_1, ..., \pi_n]|$$
$$\leq \frac{2r_{\max}\gamma}{(1-\gamma)^2}\xi + \frac{2r_{\max}}{1-\gamma}\sum_{i=1}^{n}\left[\epsilon_{m_i} \cdot \left(\sum_{k=0}^{T-1}\gamma^{k+1}\frac{|\bar{N}_i^k|}{n}\right) + \epsilon_{\pi_i} \cdot \left(\sum_{k=T}^{\infty}\gamma^{k+1}\frac{|\bar{N}_i^k|}{n}\right)\right]$$

The proof can also be found in Appendix C. Compared to Theorem 2, Corollary 1 is more general, as it is applicable to the multi-agent systems that are not fully independent. Intuitively, if a networked system seems nearly independent, local models will be effective enough. The bound indicates that when the policy in optimized in a trust region where $D(\pi, \pi_D) \leq \epsilon_{\pi_i}$, the bound would also be restricted, making monotonic update more achievable.

### 5.3 EXTENDED VALUE FUNCTION

In this section, we analyze the effect of extended value function. The idea of extended value function $V_i(s_{N_i^\kappa})$ comes from *truncated Q-function* $Q_i(s_{N_i^\kappa}, a_{N_i^\kappa})$ proposed in (Qu et al., 2020a). We prove that extended value function is an approximation of the real value function. The detailed proof of Theorem 3 is deferred to Appendix C.

**Theorem 3.** *Define $V_i(s_{N_i^\kappa}) = \mathbb{E}_{s_{N_{-i}^\kappa} \sim d_\pi}[\sum_{t=0}^{\infty} r_i^t | s_{N_i^\kappa}^0 = s_{N_i^\kappa}]$, and $V_i(s) = \mathbb{E}[\sum_{t=0}^{\infty} r_i^t | s^0 = s]$, then:*

$$|V_i(s) - V_i(s_{N_i^\kappa})| \leq \frac{r_{\max}}{1-\gamma}\gamma^\kappa.$$

From Theorem 3, it is straightforward that the global value function can be approximated with the average of all extended value functions: $|V(s) - \frac{1}{n}\sum_{i=1}^{n}V_i(s_{N_i^\kappa})| \leq \frac{r_{\max}}{1-\gamma}\gamma^\kappa$. In PPO, value functions are used for calculating advantages $\hat{A}^{(t)} = r^{(t)} + \gamma V(s^{(t+1)}) - V(s^{(t)})$, and we have proven that $V(s)$ can be estimated with the average of extended value functions $\frac{1}{n}\sum_{i=1}^{n}V_i(s_{N_i^\kappa})$. In practice, an agent might not get the value function of distant agents. However, we can prove that $\tilde{V}_i = \frac{1}{n}\sum_{j \in N_i^\kappa} V_j(s_{N_j^\kappa})$ is already very accurate for calculating the policy gradient for agent $i$. Theorem 4 justifies that the policy gradients computed based on the sum of the nearby extended value functions is a close approximation of true policy gradients.

**Theorem 4.** *Let $\hat{A}_t = r^{(t)} + \gamma V(s^{(t+1)}) - V(s^{(t)})$ be the TD residual, and $g_i = \mathbb{E}[\hat{A}\nabla_{\theta_i} \log \pi_i(a|s)]$ be the policy gradient. If $\tilde{A}_t$ and $\tilde{g}_i$ are the TD residual and policy gradient when value function $V(s)$ is replaced by $\tilde{V}_i(s) = \frac{1}{n}\sum_{j \in N_i^\kappa} V_j(s_{N_j^\kappa})$, we have:*

$$|g_i - \tilde{g}_i| \leq \frac{\gamma^{\kappa-1}}{1-\gamma}[1 - (1-\gamma^2)\frac{N_i^\kappa}{n}]r_{\max}g_{\max},$$

*where $r_{\max}$ and $g_{\max}$ denote the upper bound of the absolute value of reward and gradient, respectively.*

# 6 EXPERIMENTS

## 6.1 ENVIRONMENTS

We test our algorithm in four environments, namely Figure Eight, Ring Attenuation (Wu et al., 2017a), CACC Catchup, and CACC Slowdown (Chu et al., 2020). Detailed description and visualization of these environments is deferred to Appendix A.

**Cooperative Adaptive Cruise Control**   The objective of CACC is to adaptively coordinate a platoon of 8 vehicles to minimize the car-following headway and speed perturbations based on real-time vehicle-to-vehicle communication. CACC consists of two scenarios: Catch-up and Slow-down. In CACC Catch-up, vehicles need to catch up to the first car. In CACC Slow-down, every vehicle is faster than the optimal speed, and they need to slow down without causing any collision. The agents receives a negative reward if the headway or the speed is not optimal. Also, whenever a collision happens, a huge negative reward of -1000 is given.

**Flow environments**   This task consists of Figure Eight and Ring Attenuation. The objective of these environments is letting the automated vehicles achieve a target average speed inside the road network while avoiding collisions. The state of each vehicle is its velocity and position, and the action is the acceleration of itself. In Ring Attenuation, the objective is to achieve a high speed, while avoiding stop-and-go loops. Vehicles are rewarded with their speed, but also punished for their accelerations. In the perspective of a networked system, we assume that the vehicles are connected with the preceding and succeeding vehicle, thus resulting in a loop-structured graph.

## 6.2 BASELINES

We describe the following algorithms for performance comparison:

- CPPO: Centralized PPO learns a centralized critic $V_i(s)$. This baseline aims to analyze the performance when $\kappa$ is set to be arbitrarily huge, and is used in (Vinitsky et al., 2018) as a benchmark algorithm for networked system control.
- IC3Net (Singh et al., 2018): A communication-based multi-agent RL algorithm. The agents maintain their local hidden states with a LSTM kernel, and actively determines the communication target. Compared with DPPO, IC3Net uses hidden state and continuous communication, whereas DPPO agents directly observe the states of their neighbors.
- DPPO: Decentralized PPO learns an independent actor and critic for each agent. We implement it by using neighbor's state for extended value estimation.
- DMPO (our method): DMPO is a decentralized and model-based algorithm based on DPPO. On top of it, we use decentralized graph convolutional kernel as predictive model.

## 6.3 RESULTS

Figure 2 shows the episode reward v.s. number of training samples curves of the algorithms. We address that in CACC environments, DMPO uses decentralized SAC as base algorithm. Similar with DPPO, decentralized SAC uses extended $Q$-function $Q_i(s_{N_i^\kappa}, a_{N_i^\kappa})$ for its policy gradient. From the results, we conclude that our algorithm matches the asymptotic performance of model-free methods. It also learns the policy faster, resulting in increased sample efficiency.

The comparison between DMPO and DPPO can be viewed as an ablation study of model usage. In figure eight, DMPO increases sample efficiency at the beginning, but as the task becomes difficult, the sample efficiency of our method decreased. In a relatively easy task, ring attenuation, our method increased sample efficiency massively, compared with its model-free counterpart.

The comparison between the asymptotic performance of CPPO and DMPO or DPPO can be viewed as an ablation study of extended value function. From the result in four environments, we observe that the asymptotic performance of CPPO does not exceed that of the algorithms that uses extended value function. In this way, we conclude that by using extended value function, a centralized algorithm can be decomposed into decentralized algorithm, but the performance would not drop significantly.

Figure 3 shows the accuracy of our model in predicting the reward and state during training. The error is defined as the ratio of MSE loss to variance. From the figures, we conclude that neighborhood information is accurate enough for a model to predict the next state in these environments. However, in CACC Slow-down, local models might fail to learn the reward. We observe that the errors may increase as the agents explore new regions in the state space.

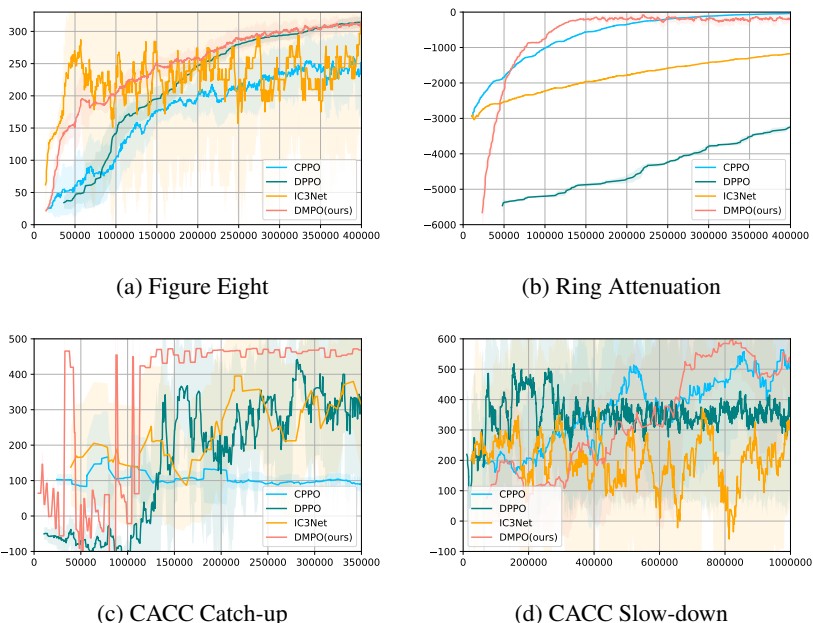

| (a) Figure Eight | (b) Ring Attenuation |
|---|---|
| (c) CACC Catch-up | (d) CACC Slow-down |

Figure 2: Training curves on multi-agent environments. Solid curves depict the mean of five trails, and shaded region correspond to standard deviation.

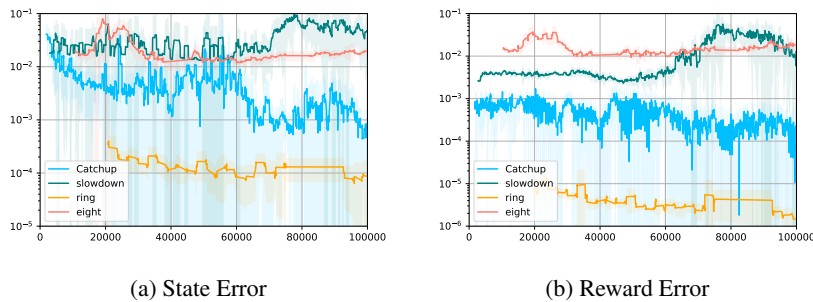

| (a) State Error | (b) Reward Error |
|---|---|

Figure 3: Figures of state and reward error. Both state error and reward error $< 10\%$ in every environment.

## 7 CONCLUSIONS

In this paper, we propose algorithm DMPO, a model-based and decentralized multi-agent RL algorithm. We then give a theoretical analysis on the algorithm to analyze its performance discrepancy, compared with a model-free algorithm. By experiments in several tasks in networked systems, we show that although our algorithm is decentralized and model-based, it matches the asymptotic performance of some state-of-art multi-agent algorithms. From the results, we also conclude that using extended value function instead of centralized value function did not sacrifice performance massively, yet it makes our algorithm scalable.

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

## A   EXPERIMENT DETAILS

The code of our algorithm can be found at `https://anonymous.4open.science/r/RL-algorithms-0E72`.

### A.1   ENVIRONMENT DESCRIPTION

The objective of CACC is to adaptively coordinate a line of vehicles to minimize the car-following headway and speed perturbations based on real-time vehicle-to-vehicle communication. We conduct experiments on two scenarios: CACC catch-up, CACC slow-down. The local observation of each agent consists of headway $h$, velocity $v$, acceleration $a$, and is shared to neighbors within two steps. The action of each agent is to choose appropriate hyper-parameters $(\alpha^\circ, \beta^\circ)$ for each OVM controller Bando et al. (1995), selected from four levels $\{(0,0),(0.5,0),(0,0.5),(0.5,0.5)\}$, where $\alpha^\circ, \beta^\circ$ denotes headway gain and relative gain for OVM controller respectively. The reward function is defined as $(h_{i,t} - h^*)^2 + (v_{i,t} - v_t^*)^2 + 0.1 a_{i,t}^2$ to punish the gap between the current state to target state and speed perturbations, where the target headway and velocity profile are $h^* = 20m$ and $v_t^*$, respectively. Whenever a collision happens ($h_{i,t} < 1m$), a large penalty of -1000 is assigned to each agent and the state becomes absorbing. An additional cost $5\left(2h_{st} - h_{i,t}\right)_+^2$ is provided in training for potential collisions. In catch-up scenario, initial headway of the first vehicle is larger than the target headway, thus the following agents learn how to catch up with the first vehicle, where target speed $v_t^* = 15m/s$ and initial headway $h_{1,0} > h_{i,0}, \forall i \neq 1$. In slow-down scenario, target speed $v_t^*$ linearly decreases to 15m/s during the first 30s and then stays at constant, thus agents learn how to slow down speed cooperatively, where initial headway $\mathrm{h}_{i,0} = \mathrm{h}^*$.

Flow environments consists of Figure Eight and Ring Attenuation. The objective of these environments is letting the automated vehicles achieve a target average speed inside the road network while avoiding collisions.

The figure eight network, previously presented in (Wu et al., 2017b), acts as a closed representation of an intersection. In a figure eight network containing a total of 14 vehicles, we witness the formation of queues resulting from vehicles arriving simultaneously at the intersection and slowing down to obey right-of-way rules. This behavior significantly reduces the average speed of vehicles in the network. The state consists of velocity and position for the vehicle. The action is the acceleration of the vehicle $a \in \mathbb{R}_{[a_{\min}, a_{\max}]}$. The objective of the learning agent is to achieve high speeds while penalizing collisions. Accordingly, the local reward function is defined as $r_i = v_{des} - |v_{des} - v_i|$, where $v_{des}$ is an arbitrary target velocity.

In Ring Attenuation, the objective is to achieve a high speed, while avoiding acceleration-deceleration loops. To achieve this, vehicles are rewarded with their speed and punished for their accelerations. The state and action of each vehicle is the same as Figure Eight. In the perspective of a networked system, we assume that the vehicles are connected with the preceding and succeeding vehicle, thus resulting in a loop-structured graph.

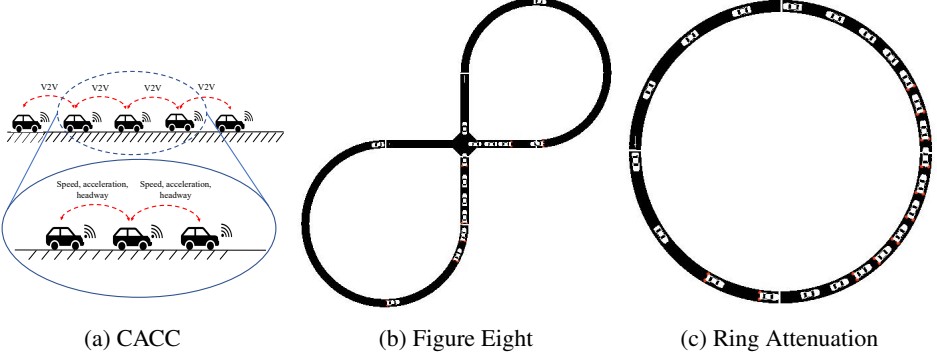

|      (a) CACC      |    (b) Figure Eight    |  (c) Ring Attenuation  |

Figure 4: Visualization of CACC and Flow environments. (a) A line of vehicles that need to keep a stable velocity and desired headway. (b) Vehicles travel in a figure eight shaped road section to learn the behavior at an intersection .(c) Vehicles travel in a ring to reduce stop-and-go waves.

## B   HYPERPARAMETERS

We list some of the key hyperparameters for DMPO and DPPO in Table 1 and 2.

| Scenario / Hyperparameter | Catch Up | Slow Down | Figure Eight | Ring Attenuation |
|---|---|---|---|---|
| Learning rate of critic | 3e-4 | 3e-4 | 5e-5 | 5e-4 |
| Learning rate of $\pi$ | 3e-4 | 3e-4 | 5e-5 | 5e-4 |
| Learning rate of Model | 3e-4 | 3e-4 | 5e-4 | 5e-4 |
| Clip | - | - | 0.15 | 0.2 |
| GAE $\lambda$ | 1 | 1 | 0.5 | 0.5 |
| KL target | - | - | 7.5e-3 | 0.01 |
| Model usage probability | 1 | 1 | 0.5 | 0.5 |
| Rollout Length | 1 | 1 | 25 | 25 |
| $\kappa$ | 1 | 1 | 3 | 3 |

Table 1: Hyperparameters for DMPO.

| Scenario / Hyperparameter | Catch Up | Slow Down | Figure Eight | Ring Attenuation |
|---|---|---|---|---|
| Learning rate of $V$ | 7e-4 | 5e-4 | 5e-4 | 1e-3 |
| Learning rate of $\pi$ | 5e-5 | 5e-5 | 5e-5 | 1e-3 |
| Clip | 0.2 | 0.2 | 0.2 | 0.2 |
| GAE $\lambda$ | 0.5 | 0.5 | 0.5 | 0.5 |
| KL target | 0.01 | 0.01 | 0.01 | 0.01 |
| $\kappa$ | 2 | 2 | 3 | 3 |

Table 2: Hyperparameters for DPPO.

## C   PROOF OF THEOREMS

### C.1   REMARKS ON PROOFS OF LEMMAS AND THEOREMS

To bound the returns, we use TV distance to assess the distance between $s_i, a_i$ at every timestep, based on the fact that $|\mathbb{E}_{s_i,a_i \sim p_1} r(s_i, a_i) - \mathbb{E}_{s_i,a_i \sim p_2} r(s_i, a_i)| \leq r_{\max} \cdot D_{TV}(p_1 \| p_2)$. To achieve this, we argue that in an INS, $s_i^T$ only depends on $s_{N_i^1}^{T-1}$, which is the 1-hop neighbors' states at previous timestep. Then, inductively, $s_i^T$ is only dependent on $s_{N_i^1}^{T-1}, s_{N_i^2}^{T-2}, ..., s_{N_i^k}^{T-k}$, and ultimately, $s_{N_i^T}^0$. Then we consider a chain of states $x^t = s_{N_i^{N-t}}, t = 0, 1, ..., N$ in Lemma 4 to analyze how $s_i^t$ depends on model errors and policy divergences in a step-by-step manner. Insipred by Janner et al., by summing the divergence at each timestep $t = 0, 1, ...,$ we can generate a overall discounted returns bound of branched rollout in Lemma 5. Lemma 6 is a simple corollary of Lemma 5, which handles normal rollout scenario.

If the returns are bounded, we can show that if every optimization step increase the return by $C$, the real return would also almost monotonically increase. Specifically, if $\hat{\eta}[\pi^{k+1}] \geq \hat{\eta}[\pi^k] + C$, combined with $\eta[\pi^{k+1}] \geq \hat{\eta}[\pi^k] - C$, we have $\eta[\pi^{k+1}] \geq \hat{\eta}[\pi^k]$. But note that $\hat{\eta}[\pi^k]$ increases monotonically, and it's a lower bound of $\eta[\pi^{k+1}]$. This means that $\hat{\eta}[\pi^{k+1}]$ would almost monotonically increase.

### C.2   PROOF OF THEOREM 1

*Proof.* $\eta^p[\pi_1, ..., \pi_n]$ and $\eta^{\hat{p}}[\pi_1, ..., \pi_n]$ differs in two ways. First, they are estimated with different transition dynamics. Second, the latter sample their states and actions with another policy, namely $\pi_D$. To deal with these divergences separately, we use an intermediate return $\eta[\pi_D]$, which is the

return of $\pi_D$ in the environment. Then the difference can be bounded by:

$$|\eta_i^p[\pi_1, ..., \pi_n] - \eta_i^{\hat{p}}[\pi_1, ..., \pi_n]| \leq \underbrace{|\eta_i^p[\pi_1, ..., \pi_n] - \eta_i^p[\pi_D]|}_{L_1} + \underbrace{|\eta_i^p[\pi_D] - \eta_i^{\hat{p}}[\pi_1, ..., \pi_n]|}_{L_2}. \quad (9)$$

To bound $L_1$, we apply Lemma 6 with $\epsilon_{m_i} = 0$. Then:

$$L_1 \leq \frac{2r_{\max}}{1 - \gamma}\Big[\epsilon_{\pi_i} + \sum_{k=0}^{\infty} \gamma^{k+1} \sum_{j \in N_i^k} \epsilon_{\pi_j}\Big]. \quad (10)$$

In $L_2$, both the dynamics and policies and different, therefore:

$$L_2 \leq \frac{2r_{\max}}{1 - \gamma}\Big[\epsilon_{\pi_i} + \sum_{k=0}^{\infty} \gamma^{k+1} \sum_{j \in N_i^k} (\epsilon_{m_j} + \epsilon_{\pi_j})\Big]. \quad (11)$$

Putting Equation 10 and 11 yields:

$$|\eta_i^p[\pi_1, ..., \pi_n] - \eta_i^{\hat{p}}[\pi_1, ..., \pi_n]| \leq \frac{2r_{\max}}{1 - \gamma}\Big[2\epsilon_{\pi_i} + \sum_{k=0}^{\infty} \gamma^{k+1} \sum_{j \in N_i^k} (\epsilon_{m_j} + 2\epsilon_{\pi_j})\Big]. \quad (12)$$

And because the binary relation of $k$-hop neighbor is symmetric, we have:

$$\frac{1}{n}\sum_{i=1}^{n} \sum_{j \in N_i^k} \epsilon_j = \sum_{j=1}^{n} \frac{|V_j^k|}{n}\epsilon_j. \quad (13)$$

Then, by averaging Equation 12, we have:

$$\begin{aligned}
&|\eta^p[\pi_1, ..., \pi_n] - \eta^{\hat{p}}[\pi_1, ..., \pi_n]| \\
&\leq \frac{1}{n}\sum_{i=1}^{n} |\eta_i^p[\pi_1, ..., \pi_n] - \eta_i^{\hat{p}}[\pi_1, ..., \pi_n]| \\
&\leq \frac{2r_{\max}}{1 - \gamma}\Big[\frac{1}{n}\sum_{i=1}^{n}\epsilon_{\pi_i} + \sum_{k=0}^{\infty}\gamma^{k+1}\frac{1}{n}\sum_{i=1}^{n}\sum_{j \in N_i^k}(\epsilon_{m_j} + 2\epsilon_{\pi_j})\Big] \\
&= \frac{2r_{\max}}{1 - \gamma}\Big[\frac{1}{n}\sum_{i=1}^{n}\epsilon_{\pi_i} + \sum_{k=0}^{\infty}\gamma^{k+1}\sum_{i=1}^{n}\frac{|N_i^k|}{n}(\epsilon_{m_i} + 2\epsilon_{\pi_i})\Big] \\
&= \frac{2r_{\max}}{1 - \gamma}\sum_{i=1}^{n}\Big[\frac{\epsilon_{\pi_i}}{n} + (\epsilon_{m_i} + 2\epsilon_{\pi_i})\cdot\sum_{k=0}^{\infty}\gamma^{k+1}\frac{|N_i^k|}{n}\Big]
\end{aligned} \quad (14)$$

$\square$

## C.3 Proof of Theorem 2

*Proof.* To bound $|\eta^p[\pi_1, ..., \pi_n] - \eta^{branch}[\pi_1, ..., \pi_n]|$, we need to analyze how do they differ from each other. $\eta^p$ denote the real returns of these policies in the environment. $\eta^{branch}$ is the returns estimated in the branched rollout scheme. To explicitly illustrate this point, we describe their difference in Table 3:

By Lemma 5, we have:

$$|\eta_i^p[\pi_1, ..., \pi_n] - \eta_i^{branch}[\pi_1, ..., \pi_n]| \leq \frac{2r_{\max}}{1 - \gamma}\Big[\sum_{k=0}^{T-1}\gamma^{k+1}\sum_{j \in N_i^k}\epsilon_{m_j} + \sum_{k=T}^{\infty}\gamma^{k+1}\sum_{j \in N_i^k}\epsilon_{\pi_j}\Big] \quad (15)$$

| branch point | before | | after | |
|---|---|---|---|---|
| | dynamics | policies | dynamics | policies |
| $\eta^p$ | $p$ | $\pi$ | $p$ | $\pi$ |
| $\eta^{branch}$ | $p$ | $\pi_D$ | $\hat{p}$ | $\pi$ |

Table 3: The difference between $\eta^p$ and $\eta^{branch}$

And because the binary relation of $k$-hop neighbor is symmetric, we have:

$$\frac{1}{n}\sum_{i=1}^{n}\sum_{j\in N_i^k}\epsilon_j = \sum_{j=1}^{n}\frac{|V_j^k|}{n}\epsilon_j. \tag{16}$$

Then, by averaging Equation 15, we have:

$$
\begin{aligned}
&|\eta^p[\pi_1,...,\pi_n] - \eta^{branch}[\pi_1,...,\pi_n]| \\
&\leq \frac{1}{n}\sum_{i=1}^{n}|\eta_i^p[\pi_1,...,\pi_n] - \eta_i^{branch}[\pi_1,...,\pi_n]| \\
&\leq \frac{2r_{\max}}{1-\gamma}\Big[\sum_{k=0}^{T-1}\gamma^{k+1}\cdot\frac{1}{n}\sum_{i=1}^{n}\sum_{j\in N_i^k}\epsilon_{m_j} + \sum_{k=T}^{\infty}\gamma^{k+1}\cdot\frac{1}{n}\sum_{i=1}^{n}\sum_{j\in N_i^k}\epsilon_{\pi_j}\Big] \\
&= \frac{2r_{\max}}{1-\gamma}\Big[\sum_{k=0}^{T-1}\gamma^{k+1}\cdot\sum_{i=1}^{n}\frac{|N_i^k|}{n}\epsilon_{m_i} + \sum_{k=T}^{\infty}\gamma^{k+1}\cdot\sum_{i=1}^{n}\frac{|N_i^k|}{n}\epsilon_{\pi_i}\Big] \\
&= \frac{2r_{\max}}{1-\gamma}\sum_{i=1}^{n}\Big[\epsilon_{m_i}\cdot\Big(\sum_{k=0}^{T-1}\gamma^{k+1}\frac{|N_i^k|}{n}\Big) + \epsilon_{\pi_i}\cdot\Big(\sum_{k=T}^{\infty}\gamma^{k+1}\frac{|N_i^k|}{n}\Big)\Big]
\end{aligned} \tag{17}
$$

$\square$

### C.4 PROOF OF COROLLARY 1

*Proof.* Although $p(s'|s,a)$ might not satisfy the criteria of an INS, we can construct another transition dynamic $\tilde{p} = \prod_{i=1}^{n}p_i(s_i|s_{N_i},a_i)$, which is the product of marginal transitions, thus being an INS. Recall that by definition, if $p$ is a transition dynamic of a $\xi$-dependent system, $\sup_{s,a}D_{TV}(p(s'|s,a)\|\tilde{p}(s'|s,a)) \leq \xi$. Then we divide the difference into two parts:

$$
\begin{aligned}
&\eta^p[\pi_1,...,\pi_n] - \eta^{branch}[\pi_1,...,\pi_n] \\
&= \underbrace{\eta^p[\pi_1,...,\pi_n] - \eta^{\tilde{p}}[\pi_1,...,\pi_n]}_{L_1} + \underbrace{\eta^{\tilde{p}}[\pi_1,...,\pi_n] - \eta^{branch}[\pi_1,...,\pi_n]}_{L_2}
\end{aligned} \tag{18}
$$

The first part is the difference of policy returns, estimated in two environments. Since there are no policy divergences, and the transition dynamics only differs in a global manner, utilizing Lemma 2 yields that:

$$D_{TV}\big(p^t(s,a)\|\tilde{p}^t(s,a)\big) \leq t\xi. \tag{19}$$

Then, we have:

$$
\begin{aligned}
L_1 &= \eta^p[\pi_1,...,\pi_n] - \eta^{\tilde{p}}[\pi_1,...,\pi_n] \\
&\leq 2r_{\max}\sum_{t=0}^{\infty}\gamma^t D_{TV}\big(p^t(s,a)\|\tilde{p}^t(s,a)\big) \\
&\leq 2r_{\max}\sum_{t=0}^{\infty}\gamma^t t\xi \\
&= \frac{2r_{\max}}{1-\gamma}\cdot\frac{\gamma}{1-\gamma}\xi.
\end{aligned} \tag{20}
$$

As for $L_2$, because $\tilde{p}$ is an INS, we can directly apply Theorem 2:

$$L_2 \leq \frac{2r_{\max}}{1-\gamma} \sum_{i=1}^{n} \left[ \epsilon_{m_i} \cdot \left( \sum_{k=0}^{T-1} \gamma^{k+1} \frac{|N_i^k|}{n} \right) + \epsilon_{\pi_i} \cdot \left( \sum_{k=T}^{\infty} \gamma^{k+1} \frac{|N_i^k|}{n} \right) \right] \tag{21}$$

Summing Equation 20 and Equation 21 completes the proof. $\square$

### C.5 PROOF OF THEOREM 3

*Proof.* In (Qu et al., 2020b), it was proven that if $s_{N_i^\kappa}$ and $a_{N_i^\kappa}$ are fixed, then no matter how other states and actions changes, $Q$-function will not change significantly:

$$|Q(s_{N_i^\kappa}, a_{N_i^\kappa}, s_{N_{-i}^\kappa}, a_{N_{-i}^\kappa}) - Q(s_{N_i^\kappa}, a_{N_i^\kappa}, s'_{N_{-i}^\kappa}, a'_{N_{-i}^\kappa})| \leq \frac{r_{\max}}{1-\gamma} \gamma^\kappa. \tag{22}$$

As value function is the expectation of Q-function

$$\begin{aligned} V_i(s) &= \mathbb{E}_{a \sim \pi} Q(s, a) \\ V_i(s_{N_i^\kappa}) &= \mathbb{E}_{a \sim \pi} Q(s_{N_i^\kappa}, a_{N_i^\kappa}), \end{aligned} \tag{23}$$

we have,

$$\begin{aligned} |V_i(s) - V_i(s_{N_i^\kappa})| &= |\mathbb{E}_{a \sim \pi} Q(s, a) - \mathbb{E}_{a \sim \pi} Q(s_{N_i^\kappa}, a_{N_i^\kappa})| \\ &\leq \mathbb{E}_{a \sim \pi} |Q(s, a) - Q(s_{N_i^\kappa}, a_{N_i^\kappa})| \\ &\leq \frac{r_{\max}}{1-\gamma} \gamma^\kappa, \end{aligned} \tag{24}$$

which concludes the proof. $\square$

### C.6 PROOF OF THEOREM 4

*Proof.* The difference of the gradients can be written as

$$\begin{aligned} g_i - \tilde{g}_i =& \mathbb{E}(\hat{A} - \tilde{A}) \nabla_{\theta_i} \log \pi_i(a_i|s_{\bar{N}_i}) \\ =& \frac{1}{n} \mathbb{E}[\sum_{j \notin N_i^\kappa} \hat{A}_j] \nabla_{\theta_i} \log \pi_i(a_i|s_{\bar{N}_i}) + \frac{1}{n} \mathbb{E}[\sum_{j \in N_i^\kappa} (\hat{A}_j - \tilde{A}_j)] \nabla_{\theta_i} \log \pi_i(a_i|s_{\bar{N}_i}) \\ =& \frac{1}{n} \mathbb{E}[\sum_{j \notin N_i^\kappa} (r_j(s_j, a_j) + \gamma V_j(s') - V_j(s))] \nabla_{\theta_i} \log \pi_i(a_i|s_{\bar{N}_i}) \\ &+ \frac{1}{n} \mathbb{E} \sum_{j \in N_i^\kappa} [(r_j(s_j, a_j) + \gamma V_j(s') - V_j(s)) - (r_j(s_j, a_j) + \gamma V_j(s'_{N_i^\kappa}) - V_j(s_{N_i^\kappa}))] \nabla_{\theta_i} \log \pi_i(a_i|s_{\bar{N}_i}) \\ =& L_1 + L_2. \end{aligned} \tag{25}$$

Because for any function $b(s)$ that depends only on $s$, $\mathbb{E}[b(s) \log \pi_i^{\theta_i}(a_i|s_{\bar{N}_i})] = 0$. Therefore, $L_2$ in Equation 25 becomes:

$$\begin{aligned} |L_2| \leq& \frac{1}{n} \mathbb{E} \sum_{j \in N_i^\kappa} \gamma |V_j(s') - V_j(s'_{N_i^\kappa})| ||\nabla_{\theta_i} \log \pi_i(a_i|s_{\bar{N}_i})| \\ \leq& \frac{|N_i^\kappa|}{n} \frac{\gamma^{\kappa+1}}{1-\gamma} r_{\max} g_{\max}. \end{aligned} \tag{26}$$

For $L_1$, note that $r_j(s_j, a_j) + \gamma V_j(s') - V_j(s) = -V_j(s) + r_j(s_j, a_j) + \sum_{t=1}^{\kappa-2} \mathbb{E} \gamma^t r_j(s_j^t, a_j^t) + \gamma^{\kappa-1} V_j(s^{\kappa-1})$. And in an INS, $s_j^t, a_j^t, t = 0, 1, ..., \kappa - 2$ is not affected by policy $\pi_i$ if $j \notin N_i^\kappa$, we have that

$$\begin{aligned} |L_1| \leq& \frac{1}{n} \mathbb{E} \sum_{j \notin N_i^\kappa} |\gamma^{\kappa-1} V_j(s^{\kappa-1})| ||\nabla_{\theta_i} \log \pi_i(a_i|s_{\bar{N}_i})| \\ \leq& (1 - \frac{N_i^\kappa}{n}) \frac{\gamma^{\kappa-1}}{1-\gamma} r_{\max} g_{\max}. \end{aligned} \tag{27}$$

Put Equation 26 and 27 together, we have

$$
\begin{aligned}
|g_i - \tilde{g}_i| \leq& |L_1| + |L_2| \\
\leq& \frac{|N_i^\kappa|}{n} \frac{\gamma^{\kappa+1}}{1-\gamma} r_{\max} g_{\max} + (1 - \frac{N_i^\kappa}{n}) \frac{\gamma^{\kappa-1}}{1-\gamma} r_{\max} g_{\max} \\
=& \frac{\gamma^{\kappa-1}}{1-\gamma} [1 - (1 - \gamma^2) \frac{N_i^\kappa}{n}] r_{\max} g_{\max}.
\end{aligned}
\tag{28}
$$

$\square$

## D  USEFUL LEMMAS

**Lemma 1.** *(TVD of Joint Distribution) Consider two distributions $p_1(x,y) = p_1(x)p_1(y|x)$ and $p_2(x,y) = p_2(x)p_2(y|x)$. The total variation distance between them can be bounded as:*

$$
\begin{aligned}
D_{TV}\big(p_1(x,y)\|p_2(x,y)\big) &\leq D_{TV}\big(p_1(x)\|p_2(x)\big) + \mathbb{E}_{x\sim p_2}\big[D_{TV}\big(p_1(y|x)\|p_2(y|x)\big)\big] \\
&\leq D_{TV}\big(p_1(x)\|p_2(x)\big) + \max_x D_{TV}\big(p_1(y|x)\|p_2(y|x)\big)
\end{aligned}
$$

*Proof.*

$$
\begin{aligned}
D_{TV}\big(p_1(x,y)\|p_2(x,y)\big) &= \frac{1}{2}\sum_{x,y} |p_1(x,y) - p_2(x,y)| \\
&= \frac{1}{2}\sum_{x,y} |p_1(x)p_1(y|x) - p_2(x)p_2(y|x)| \\
&= \frac{1}{2}\sum_{x,y} |p_1(x)p_1(y|x) - p_2(x)p_1(y|x) + p_2(x)p_1(y|x) - p_2(x)p_2(y|x)| \\
&\leq \frac{1}{2}\sum_{x,y} |p_1(x) - p_2(x)|p_1(y|x) + \frac{1}{2}\sum_{x,y} p_2(x)|p_1(y|x) - p_2(y|x)| \\
&= \frac{1}{2}\sum_{x} |p_1(x) - p_2(x)| + \sum_{x} p_2(x) D_{TV}(p_1(y|x)\|p_2(y|x)) \\
&= D_{TV}\big(p_1(x)\|p_2(x)\big) + \mathbb{E}_{x\sim p_2}\big[D_{TV}\big(p_1(y|x)\|p_2(y|x)\big)\big] \\
&\leq D_{TV}\big(p_1(x)\|p_2(x)\big) + \max_x D_{TV}\big(p_1(y|x)\|p_2(y|x)\big)
\end{aligned}
$$

$\square$

**Lemma 2.** *Suppose there are two chains of distributions $\{x_1^t, t \geq 0\}, \{x_2^t, t \geq 0\}$. At time $t$, the states of both chains share an identical state space $x_1^t, x_2^t \in \mathcal{X}^t$. Suppose these two chains satisfy a Markov-like property: $p^{t+1}(x^{t+1}|x^t, ..., x^1, x^0) = p^{t+1}(x^{t+1}|x^t)$. Then, the TVD of distributions of two chains at time $t$ can be decomposed as:*

$$
D_{TV}\big(p_1^T(x^T)\|p_2^T(x^T)\big) \leq D_{TV}\big(p_1^0(x^0)\|p_2^0(x^0)\big) + \sum_{t=1}^{T} \mathbb{E}_{s^{t-1}\sim p_2^{t-1}} D_{TV}\big(p_1^t(x^t|x^{t-1})\|p_2^t(x^t|x^{t-1})\big).
$$

*Proof.* We prove this lemma by induction.

When $T = 0$, it's easy to see that this lemma is true.

Assume it is true for $T = k$. We have:

$$|p_1^{k+1}(x^{k+1}) - p_2^{k+1}(x^{k+1})|$$
$$= |\sum_{x^k} [p_1^{k+1}(x^{k+1}|x^k)p_1^k(x^k) - p_2^{k+1}(x^{k+1}|x^k)p_2^k(x^k)]|$$
$$\leq \sum_{x^k} |p_1^{k+1}(x^{k+1}|x^k)p_1^k(x^k) - p_2^{k+1}(x^{k+1}|x^k)p_2^k(x^k)|$$
$$= \sum_{x^k} |p_1^{k+1}(x^{k+1}|x^k)p_1^k(x^k) - p_1^{k+1}(x^{k+1}|x^k)p_2^k(x^k)$$
$$+ p_1^{k+1}(x^{k+1}|x^k)p_2^k(x^k) - p_2^{k+1}(x^{k+1}|x^k)p_2^k(x^k)|$$
$$\leq \sum_{x^k} \left[p_2^k(x^k)|p_1^{k+1}(x^{k+1}|x^k) - p_2^{k+1}(x^{k+1}|x^k)| + p_1^{k+1}(x^{k+1}|x^k)|p_1^k(x^k) - p_2^k(x^k)|\right]$$
$$= \mathbb{E}_{x^k \sim p_2^k}[|p_1^{k+1}(x^{k+1}|x^k) - p_2^{k+1}(x^{k+1}|x^k)|] + \sum_{x^k} p_1^{k+1}(x^{k+1}|x^k)|p_1^k(x^k) - p_2^k(x^k)|$$

$$D_{TV}\left(p_1^{k+1}(x^{k+1})\|p_2^{k+1}(x^{k+1})\right)$$
$$= \frac{1}{2} \sum_{x^{k+1}} |p_1^{k+1}(x^{k+1}) - p_2^{k+1}(x^{k+1})|$$
$$\leq \frac{1}{2} \sum_{x^{k+1}} \left(\mathbb{E}_{x^k \sim p_2^k}[|p_1^{k+1}(x^{k+1}|x^k) - p_2^{k+1}(x^{k+1}|x^k)|]\right.$$
$$+ \sum_{x^k} p_1^{k+1}(x^{k+1}|x^k)|p_1^k(x^k) - p_2^k(x^k)| \Bigg)$$
$$= \mathbb{E}_{s^k \sim p_2^k} D_{TV}\left(p_1^{k+1}(x^{k+1}|x^k)\|p_2^{k+1}(x^{k+1}|x^k)\right) + D_{TV}\left(p_1^k(x^k)\|p_2^k(x^k)\right)$$
$$\leq D_{TV}\left(p_1^0(x^0)\|p_2^0(x^0)\right) + \sum_{t=1}^{k+1} \mathbb{E}_{s^{t-1} \sim p_2^{t-1}} D_{TV}\left(p_1^t(x^t|x^{t-1})\|p_2^t(x^t|x^{t-1})\right)$$

Then this theorem holds for all $n \in \mathbb{N}$. $\qquad\square$

**Lemma 3.** *Consider two distributions with pdf/pmf $p(x)$ and $q(x)$, where $x = (x_1, ..., x_n) \in \mathbb{R}^n$. Suppose $p$ and $q$ can be factorized as: $p(x) = \prod_{i=1}^n p_i(x_i), q(x) = \prod_{i=1}^n q_i(x_i)$. Then we have:*

$$D_{TV}[p(x)\|q(x)] \leq \sum_{i=1}^n D_{TV}[p_i(x_i)\|q_i(x_i)].$$

*Also, if the distance is measured by KL-divergence, we have:*

$$D_{KL}[p(x)\|q(x)] = \sum_{i=1}^n D_{KL}[p_i(x_i)\|q_i(x_i)].$$

*Proof.* We prove this result for discrete distributions, yet by replacing sum with integration, this result stays true in continuous case.

$$
\begin{aligned}
D_{TV}[p(x)\|q(x)] &= \frac{1}{2}\sum_x |p(x) - q(x)| \\
&= \frac{1}{2}\sum_{x_1,\ldots,x_n} |\prod_{k=1}^n p_k(x_k) - \prod_{k=1}^n q_k(x_k)| \\
&= \frac{1}{2}\sum_{x_1,\ldots,x_n} \Big| \sum_{i=1}^n \Big[ \prod_{k=1}^{i-1} p_k(x_k) \prod_{k=i+1}^n q_k(x_k)(p_i(x_i) - q_i(x_i)) \Big] \Big| \\
&\leq \frac{1}{2}\sum_{x_1,\ldots,x_n} \sum_{i=1}^n \Big[ \prod_{k=1}^{i-1} p_k(x_k) \prod_{k=i+1}^n q_k(x_k)|p_i(x_i) - q_i(x_i)| \Big] \\
&= \frac{1}{2}\sum_{i=1}^n \sum_{x_i} |p_i(x_i) - q_i(x_i)| \\
&= \sum_{i=1}^n D_{TV}[p_i(x_i)\|q_i(x_i)].
\end{aligned}
$$

In KL-divergence case, we have:

$$
\begin{aligned}
D_{KL}[p(x)\|q(x)] &= \sum_x p(x)\log\frac{p(x)}{q(x)} \\
&= \sum_{x_1,\ldots,x_n} \Big[ p(x_1,\ldots,x_n) \sum_{i=1}^n \log\frac{p_i(x_i)}{q_i(x_i)} \Big] \\
&= \sum_{i=1}^n \sum_{x_1,\ldots,x_n} \Big[ \prod_{k=1}^n p_k(x_k)\log\frac{p_i(x_i)}{q_i(x_i)} \Big] \\
&= \sum_{i=1}^n \sum_{x_i} p_i(x_i)\log\frac{p_i(x_i)}{q_i(x_i)} \\
&= \sum_{i=1}^n D_{KL}(p_i(x_i)\|q_i(x_i)).
\end{aligned}
$$

$\square$

**Lemma 4.** *(N-step distribution distance) Suppose the expected TVD between two dynamics transitions is bounded as $\epsilon_{m_i} = \max_{s_{N_i},a_i} D_{TV}\big[p(s_i'|s_{N_i},a_i)\|\hat{p}(s_i'|s_{N_i},a_i)\big]$ and $\epsilon_{\pi_i} = \max_s D_{TV}\big[\pi_i(a_i|s_{N_i})\|\hat{\pi}_i(a_i|s_{N_i})\big]$. If $N \leq \kappa$, the N-step distribution distance is bounded as:*

$$
D_{TV}\big[p_i^N(s_i)\|\hat{p}_i^N(s_i)\big] \leq \sum_{t=0}^{N-1}\sum_{j\in N_i^t}(\epsilon_{\pi_j} + \epsilon_{m_j}).
$$

*Thus,*

$$
D_{TV}\big[p_i^N(s_i,a_i)\|\hat{p}_i^N(s_i,a_i)\big] \leq \epsilon_{\pi_i} + \sum_{t=0}^{N-1}\sum_{j\in N_i^t}(\epsilon_{\pi_j} + \epsilon_{m_j}).
$$

*Proof.* Consider a chain of regional state $x^t = s_{N_i^{N-t}}$. Two chains of distributions $p^t(x^t), \hat{p}(x^t)$ denote the distributions of $x^t$ under the environment and our model, respectively. First, because of the property of an INS, these two chains' transition dynamics can be decomposed as:

$$p(x^t|x^{t-1}) = p(s'_{N_i^{N-t}}|s_{N_i^{N-t+1}})$$
$$= \prod_{j \in N_i^{N-t}} p_j(s'_j|s_{V_j}).$$

And because $p_j(s'_j|s_{V_j}) = \sum_{a_j} p_j(s'_j|s_{V_j}, a_j)\pi_j(a_j|s_{V_j})$, by the property of TVD, we know that:

$$D_{TV}\left[p_j(s'_j|s_{V_j})\|\hat{p}_j(s'_j|s_{V_j})\right] \leq \epsilon_{\pi_j} + \epsilon_{m_j}.$$

With Lemma 3, we have:

$$D_{TV}\left[p(x^t|x^{t-1})\|\hat{p}(x^t|x^{t-1})\right] \leq \sum_{j \in N_i^{N-t}} (\epsilon_{\pi_j} + \epsilon_{m_j}).$$

Then, by Lemma 2, we know that

$$D_{TV}\left[p_i^N(s_i)\|\hat{p}_i^N(s_i)\right] = D_{TV}\left[p_i^N(x^N)\|\hat{p}_i^N(x^N)\right]$$
$$\leq \sum_{t=1}^{N} \sum_{j \in N_i^{N-t}} (\epsilon_{\pi_j} + \epsilon_{m_j})$$
$$= \sum_{t=0}^{N-1} \sum_{j \in N_i^t} (\epsilon_{\pi_j} + \epsilon_{m_j}),$$

which completes the proof of the first part. And by Lemma 1, the second part holds true. □

**Lemma 5.** *(Returns bound measured in branched rollout) Consider two MDPs $p(s)$ and $\hat{p}(s)$. Suppose they both adopt $T$-branched rollout scheme. Before the branch, suppose the dynamics distributions are bounded as $\max_{s_{N_i}, a_i} D_{TV}\left(p_i^{pre}(s'_i|s_{N_i}, a_i)\|\hat{p}_i^{pre}(s'_i|s_{N_i}, a_i)\right) = \epsilon_{m_i}^{pre}$, and policy divergences are bounded as $\max_{s_{N_i}} D_{TV}\left(\pi_i^{pre}(a_i|s_{N_i})\|\hat{\pi}_i^{pre}(a_i|s_{N_i})\right) = \epsilon_{\pi_i}^{pre}$. After the branch, $\epsilon_{m_i}^{post}$ and $\epsilon_{\pi_i}^{post}$ are defined similarly. Then the (local) $T$-step returns are bounded as:*

$$|\eta_i - \hat{\eta}_i| \leq \frac{2r_{\max}}{1-\gamma}\left[\epsilon_{\pi_i}^{post} + \sum_{k=0}^{T-1} \gamma^{k+1} \sum_{j \in N_i^k} (\epsilon_{m_j}^{post} + \epsilon_{\pi_j}^{post}) + \sum_{k=T}^{\infty} \gamma^{k+1} \sum_{j \in N_i^k} (\epsilon_{m_j}^{pre} + \epsilon_{\pi_j}^{pre})\right]$$

*Proof.* We prove this by estimating the state-action distribution divergence at each timestep. For notation simplicity, we denote $\epsilon_{N_i^k}^{pre} = \epsilon_{N_i^k}^{pre} = \sum_{j \in N_i^k}(\epsilon_{m_j}^{pre} + \epsilon_{\pi_j}^{pre})$, and $\epsilon_{N_i^k}^{post}$ analogously. If we divide the chains into two parts: pre-branch and post-branch, and apply Lemma 4 on both parts, we have:

For $t \leq T$:

$$D_{TV}\left[p_i^t(s_i, a_i)\|\hat{p}_i^t(s_i, a_i)\right] \leq \epsilon_{\pi_i}^{post} + \sum_{k=0}^{t-1} \epsilon_{N_i^k}^{post},$$

and for $t > T$:

$$D_{TV}\left[p_i^t(s_i, a_i)\|\hat{p}_i^t(s_i, a_i)\right] \leq \epsilon_{\pi_i}^{post} + \sum_{k=0}^{T-1} \epsilon_{N_i^k}^{post} + \sum_{k=T}^{t-1} \epsilon_{N_i^k}^{pre}.$$

Then, the difference of discounted distribution can be bounded as:

$$D_{TV}\big[p_i(s_i,a_i)\|\hat{p}_i(s_i,a_i)\big] \leq (1-\gamma)\sum_{t=0}^{\infty} D_{TV}\big[p_i^t(s_i,a_i)\|\hat{p}_i^t(s_i,a_i)\big]$$

$$\leq (1-\gamma)\sum_{t=0}^{T}\gamma^t\big(\epsilon_{\pi_i}^{post}+\sum_{k=0}^{t-1}\epsilon_{N_i^k}^{post}\big)+$$

$$(1-\gamma)\sum_{t=T+1}^{\infty}\gamma^t\big(\epsilon_{\pi_i}^{post}+\sum_{k=0}^{T-1}\epsilon_{N_i^k}^{post}+\sum_{k=T}^{t-1}\epsilon_{N_i^k}^{pre}\big)$$

$$= \epsilon_{\pi_i}^{post}+(1-\gamma)\sum_{k=0}^{T-1}\epsilon_{N_i^k}^{post}(\gamma^{k+1}+\gamma^{k+2}+...)+$$

$$(1-\gamma)\sum_{k=T}^{\infty}\epsilon_{N_i^k}^{pre}(\gamma^{k+1}+\gamma^{k+2}+...)$$

$$= \epsilon_{\pi_i}^{post}+\sum_{k=0}^{T-1}\epsilon_{N_i^k}^{post}\gamma^{k+1}+\sum_{k=T}^{\infty}\epsilon_{N_i^k}^{pre}\gamma^{k+1}.$$

We can convert this bound into the returns bound:

$$|\eta_i-\hat{\eta}_i| \leq \sum_{t=0}^{\infty}\gamma^t|r_i(s_i^t,a_i)-r_i(\hat{s}_i^t,\hat{a}_i^t)|$$

$$\leq \frac{2r_{\max}}{1-\gamma}(1-\gamma)\sum_{t=0}^{\infty} D_{TV}\big[p_i^t(s_i,a_i)\|\hat{p}_i^t(s_i,a_i)\big]$$

$$= \frac{2r_{\max}}{1-\gamma}D_{TV}\big[p_i(s_i,a_i)\|\hat{p}_i(s_i,a_i)\big]$$

$$= \frac{2r_{\max}}{1-\gamma}\big[\epsilon_{\pi_i}^{post}+\sum_{k=0}^{T-1}\epsilon_{N_i^k}^{post}\gamma^{k+1}+\sum_{k=T}^{\infty}\epsilon_{N_i^k}^{pre}\gamma^{k+1}\big]$$

$\square$

**Lemma 6.** *(Returns bound measured in full length rollout) Consider two MDPs $p(s)$ and $\hat{p}(s)$. Suppose they both run their rollouts until the end of every trajectory, and the dynamic distributions are bounded as $\max_{s_{N_i},a_i} D_{TV}\big(p_i(s_i'|s_{N_i},a_i)\|\hat{p}_i(s_i'|s_{N_i},a_i)\big)=\epsilon_{m_i}$, while policy divergences are bounded as $\max_{s_{N_i}} D_{TV}\big(\pi_i(a_i|s_{N_i})\|\hat{\pi}_i(a_i|s_{N_i})\big)=\epsilon_{\pi_i}$. Then the (local) returns are bounded as:*

$$|\eta_i-\hat{\eta}_i| \leq \frac{2r_{\max}}{1-\gamma}\big[\epsilon_{\pi_i}+\sum_{k=0}^{\infty}\gamma^{k+1}\sum_{j\in N_i^k}(\epsilon_{m_j}+\epsilon_{\pi_j})\big]$$

*Proof.* We can think it as a special case of branched rollout, where $\epsilon^{post}=\epsilon^{pre}=\epsilon$ for every subscript, and $T=0$. From this perspective, applying the result of Lemma 5 completes the proof.
$\square$

