# OpenReview forum: "Fully Decentralized Model-based Policy Optimization with Networked Agents"
_ICLR.cc/2022/Conference — ICLR 2022 Submitted_

### Official Review · Reviewer_QuxM · 2021-11-02

**Correctness:** 3
**Technical Novelty And Significance:** 2
**Empirical Novelty And Significance:** 2
**Recommendation:** 5
**Confidence:** 3

**Main Review:**

Q1. The current study assumes that the target system is an independent networked system or \ksi-dependent system. First of all, the used assumptions sound to be too strong to model realistic multi-agent systems. For example, the assumption that the reward function is only dependent on local state s_i and the local action a_i makes the entire system to be interacting through the joint state transition rather than interaction through the reward. Furthermore, the transaction of an individual state only depends on the action of the corresponding agent. Would you please discuss the problems or characteristics of the target system where the proposed assumptions hold?

Q2. The proposed assumptions are held in the target problems (environments) used in this study? What are the common aspects of these environments, and why are these environments particularly selected? In other words, the used environments are representative environments where the proposed method can work nicely? As the authors mentioned, there is no single MARL algorithm that can solve all different problems. Therefore, it would be nice if the authors describe the characteristics or examples of the target problem where the proposed assumption and the proposed method work nicely.

Q3. The idea of model-based MARL is loosely explained. Section 4.1 discusses constructing a local state transition model using GCN, and section 4.2 discusses the PPO algorithm with a truncated value function. First of all, the paper does not explain how to use the learned local dynamic model is used in training the value function and policy. Although the procedure is loosely explained in Algorithm 1, the notations used in Algorithm 1 are not fully explained. Furthermore, if the trained model is used to generate sample trajectory for model-free MARL, the proposed method's algorithmic novelty in developing model-based MARL is very limited. There should be more diversified ways to use the learned dynamic models to learn decentralized policy better. The author should investigate these approaches.
Q4. The dynamic models in MARL are generally composed of (1) environmental transition given the current joint state and the joint action and (2) the behavior models predicting other agents’ action given the current state. The current study only focused on a limited version of (1) while ignoring (2).


Q5. What are the relationships between sections 5.2 and 5.3? Each section seems to discuss independent analysis. For example, 5.2 discusses return bound when performing rollout, while section 5.3 discusses error bound for the truncated value function and policy gradient. Are these results are an extension of the previous study? If yes, what are the new aspects? In addition, how can these analyses be used to prove the effectiveness of the proposed model-based MARL? Is there any chance that these analysis results can be integrated and used to show that the proposed independent networked system assumption can effectively solve the target problem? Please elaborate on the significance and meaning of each theorem in terms of how the proposed method solves the target problem.


Q6. The assumption used for modeling the independent dynamic model is not used for estimating the truncated value function.


Q7. Experiment results are not sufficient to validate the effectiveness of the proposed method. Particularly, the proposed method does not excel other baselines when the target problem is complicated. The model-based approach should be beneficial, especially for complex problems. In addition, what is the intention of comparing the approximation errors for different environment models?


**Summary Of The Paper:**

This paper proposes a decentralized model-based reinforcement learning algorithm for networked multi-agent systems where cooperative agents communicate locally with their neighbors.

**Summary Of The Review:**

Although the current study discusses a vital topic in MARL, the current study focuses very limited class of MARL with quite strong assumptions. The theoretical analysis can be possibly considered novelties for the current study; however, the importance and the meanings of the theoretical analysis are not fully explained in the present paper. Finally, the experiment results are too limited to validate the effectiveness of the proposed method.

---

> ### Author Response · Authors · 2021-11-18
> **Thanks for your constructive comments. We would like to further explain our work here.**
>
>
>
>
> A1&A2: This assumption is an extension of those in [1]. Intuitively, many networked environments possess some extent of locality, that local states and actions would not affect distant states immediately. $\xi$-dependent systems is an abstraction of such property. Though this may not apply to some environments, this kind of abstraction makes the result of theoretical analysis elegant. The properties of $\xi$-dependent system might be further investigated in future works. When choosing the environments on which DMPO is applied, we suggest case-by-case analysis. The environments that we choose are networked, and also has some kind of locality. Take the environment of Ring Attenuation as an example. The velocity and position of an vehicle is mostly determined by the action of itself. And based on the experience of driving in real world, the information of nearby vehicles is enough to give an optimal policy.
>
>
>
> A3: It is true that some of the notations in Algorithm 1 is not explicitly explained, such as $\mathcal{D}^{model}$.  We have updated our manuscript to better explain the notations. Our contribution in terms of model-based RL is that we proved that branched rollout, which was initially proposed in [2], would result in decreased performance discrepancy, especially in networked systems. This is explained in Theorem 2.
>
>
>
> A4: There are indeed some model-based RL works that focus on predicting the action of others. However, we did not include that in our assumptions because this would introduce instability in the objective of local models, because the actions of other agents changes along with model training. We would leave the analysis of such models to future works.
>
>
>
> A5:  Thank you for a great suggestion. Theorems in 5.2 is about how would a model affect the returns, and those in 5.3 is about the effect of extended value function. These sections cannot be integrated in principle, and here's why:
>
> The objective of each agent should be (in cooperative environments) $J=\mathbb{E}\sum_{i\in\{1,...,n\}}V_i(s)$. However, in our assumption, agent $i$ optimize $J_i=\mathbb{E}\sum_{j\in N_i^\kappa}V_j(s_{N_i^\kappa})$, due to the cost of communication.  Therefore the objective varies for every agent. Section 5.3 gives an intuition that these objectives' gradients are good approximators of real gradient, meaning $\nabla_{\theta_i}J_i\approx\nabla_{\theta_i}J$. But the bound of gradients cannot be converted to the bound of performance.
>
> But actually, completely decentralized RL is even harder to analyze, and in section 5.3, we show that extended value function is better than normal decentralized RL algorithms. To understand this, think that $\kappa=0$ is the scenario of decentralized RL without communication. This should be considered as the significance of our work.
>
>
>
> A6: As an extension of the theorems in [1], this assumes an INS.
>
>
>
> A7: The intention of comparing the approximation errors is to show how well the localized models are. Theoretically, localized models should perform better in more independent environments. Therefore, we assume that the error of the models should reflect some information of the environment.
>
>
>
> [1] Qu, Guannan, Adam Wierman, and Na Li. "Scalable reinforcement learning of localized policies for multi-agent networked systems." *Learning for Dynamics and Control*. PMLR, 2020. http://arxiv.org/abs/1912.02906
>
> [2] Janner, Michael, et al. "When to Trust Your Model: Model-Based Policy Optimization." *Advances in Neural Information Processing Systems* 32 (2019): 12519-12530. https://arxiv.org/abs/1906.08253

---

> > ### Comment · Reviewer_QuxM · 2021-11-21
> > **Thank you for your responses.**
> >
> > Thanks for the detailed reply. If the authors present the theoretical analysis on the performance of the proposed model-based MARL by combining Theorem 5.2 and 5.3, it will be a very impressive paper. I have increased my score, but the score is not yet above the acceptance threshold because I believe there is more room to improve this paper, especially the experiment section and method description.

---

> > > ### Author Response · Authors · 2021-11-21
> > > **Thank you, and we've updated our paper to explain the method better.**
> > >
> > > Thank you for your comment. We have updated our manuscript to better explain the notations. We would also kindly remind you to update the score to reflect your recommendation.

---

### Official Review · Reviewer_2Aak · 2021-11-03

**Correctness:** 4
**Technical Novelty And Significance:** 2
**Empirical Novelty And Significance:** 2
**Recommendation:** 5
**Confidence:** 4

**Main Review:**

It is certainly necessary to have decentralized multi-agent reinforcement algorithm to solve problems such as autonomous driving, wireless communications and multi-player games. There are a few places I believe that the authors can significantly improve their paper.

1) The insight and essence of the DMPO algorithm are not adequately addressed. For the three key components: localized model, policy with one-step communication, and extended value function, other than the extended value function the purpose of the other two are not really explained well. As a consequence, Sec. 4.1 and 4.2 really very difficult to understand. Also, a complexity analysis of the algorithm is very desirable.
2) As pointed out by the authors, the effective of theoretical bounds depends on the selection of the discount factor. This limits the selection of the rollout T, since when T become large, then the advantage of the expressions in Theorem 2 will go down significantly; This also limits the the effective of Theorem 3.
#) In the numerical experiments,DMPO does not stand out as a significant improvement, maybe more experiments can be conduct to demonstrate the effectiveness of the algorithm.

**Summary Of The Paper:**

The authors proposed a decentralized multi-agent reinforcement learning algorithm, provide theoretical bounds on its performance, and conducted numerical experiments on its performance.

**Summary Of The Review:**

Overall, both the theoretical results and experiments can be strengthened to make a better case for the proposed algorithm.

---

> ### Author Response · Authors · 2021-11-18
> **Thanks for your constructive comments. We would like to further explain our work here.**
>
>
> Q1: For the three key components: localized model, policy with one-step communication, and extended value function, other than the extended value function the purpose of the other two are not really explained well.
>
>
>
> A1: The reason behind localized model and policy with one-step communication is decentralization. When multi-agent algorithms are applied to large environments where the number of agents is high, centralized algorithms would be unscalable. Reasons include communicational lag, dimensional explosion, heterogeneous agents, etc. Localized model and one-step policy is not only a component, but rather a must. Our work is focused on analyzing the effect of these characteristics theoretically and empirically.
>
>
>
> Q2:  As pointed out by the authors, the effective of theoretical bounds depends on the selection of the discount factor. This limits the selection of the rollout T, since when T become large, then the advantage of the expressions in Theorem 2 will go down significantly; This also limits the the effective of Theorem 3.
>
>
>
> A2: This is actually a misinterpretation of our theoretical analysis. In Theorem 2, the bound of performance is the sum of
> $
> \epsilon_{m_i}(\sum_{k=0}^{T-1}\gamma^{k+1}\frac{|N_i^k|}{n})+\epsilon_{\pi_i}(\sum_{k=T}^\infty\gamma^{k+1}\frac{|N_i^k|}{n})
> $
> . When the model is fairly accurate, meaning $\epsilon_{m_i}<\epsilon_{\pi_i}$, the bound would actually decrease as $T$ increase. Intuitively, when the model becomes accurate, the bottleneck of inaccuracy of policy update is not model inaccuracy, but the policy divergence allowed by the algorithm, $\epsilon_\pi$. Therefore, the length of rollout, $T$, can increase to augment the performance of the algorithm.

---

### Official Review · Reviewer_mC6k · 2021-11-03

**Correctness:** 4
**Technical Novelty And Significance:** 2
**Empirical Novelty And Significance:** 1
**Recommendation:** 5
**Confidence:** 3

**Main Review:**

Strengths: the theoretical analysis is solid.(the reviewer did not check the correctness).

Weaknesses:
1.	This paper incorporates several ideas to develop a decentralized RL algorithm. Compared with previous works on centralized, model-free, and tabular settings, this work investigates decentralized, model based, and deep learning model settings. However, from each single viewpoint, from centralized to decentralized, for instance, the problem has been addressed in prior literature. This makes the present study seem to be an ensemble of well-developed techniques. In this context, the authors are expected to justify their contributions by showing that the theoretical analysis is not a trivial integration of existing analysis schemes, if it is not. In other words, please clearly state the challenges and novelty of the paper.
2.	The improvement of sampling efficiency comes at the cost of training predictive models. Do the sampling savings compensate for the training cost? Is it possible to show that the learning process with the proposed model-based approach is more efficient in terms of time?
3.	As pointed out in the empirical study section, the predictive model might fail to learning the reward for CACC slow. What is the reason? Does this imply the proposed model-based scheme is not stable in some circumstances. The robustness of the method is also not verified. Maybe including more complicated RL tasks in the experiments will make the work more convincing and solid.
4.	Typos:
abstract: exponential -> exponentially,
Paragraph 1 on page 3, Each agent possess a localized …, possesses
Paragraph 1 on page 3, … reward functions is …, are
Last paragraph on page 4, Independent RL algorithms that observes only… , observe



**Summary Of The Paper:**

This paper proposes a model-based decentralized RL algorithm. Theoretical analysis is presented, and the theoretical results are validated with empirical studies on several vehicle cooperative cruise control tasks.

**Summary Of The Review:**

The novelty and the contributions in theory are not obviously presented.

---

> ### Author Response · Authors · 2021-11-18
> **We thank for the valuable reviews and provide answers to the questions.**
>
>
> Q1: However, from each single viewpoint, from centralized to decentralized, for instance, the problem has been addressed in prior literature. This makes the present study seem to be an ensemble of well-developed techniques. In this context, the authors are expected to justify their contributions by showing that the theoretical analysis is not a trivial integration of existing analysis schemes, if it is not.
>
>
>
> A1: There are well-developed analysis scheme on single-agent model-based RL, and it can apply to multi-agent scenarios as well. The direct application of single-agent analysis would result in a bound like $\frac{2r_{\max}}{1-\gamma}\sum_{i=1}^n[\frac{\epsilon_{\pi_i}}{n}+(\epsilon_{m_i}+2\epsilon_{\pi_i})\cdot\frac{1}{1-\gamma}]$。Our contribution of theoretical analysis is that in large environments where distant information do not affect local transition, this bound can be reduced to $\frac{2r_{\max}}{1-\gamma}\sum_{i=1}^n[\frac{\epsilon_{\pi_i}}{n}+(\epsilon_{m_i}+2\epsilon_{\pi_i})\cdot\sum_{k=0}^\infty\gamma^{k+1}\frac{|N_i^\kappa|}{n}]$ (Theorem 1). The difference is that in the last term, $\frac{1}{1-\gamma}$ is reduced to $\sum_{k=0}^\infty\gamma^{k+1}\frac{|N_i^\kappa|}{n}$, and within a large environment where $n$, the number of agents, is large, $|N_i^\kappa|/n$ is a small amount for smaller $k$'s. This suggest that localized model is relatively better in larger environments.
>
>
>
> Q2: Do the sampling savings compensate for the training cost? Is it possible to show that the learning process with the proposed model-based approach is more efficient in terms of time?
>
>
>
> A2: The usage of predictive model would inevitably introduce more computational cost, resulting in increased wall-clock training time. This also applies for other model-based algorithms. SimPLe in [1] discussed about this issue in its paper. This phenomenon is specifically shown in [2], where the clock-time to reach 200k timestep of model-based algorithms is always higher than model-free counterparts. However, sample efficiency is crucial in other ways, especially when data collection is expensive.
>
>
>
> Q3: As pointed out in the empirical study section, the predictive model might fail to learning the reward for CACC slow. What is the reason? Does this imply the proposed model-based scheme is not stable in some circumstances. The robustness of the method is also not verified. Maybe including more complicated RL tasks in the experiments will make the work more convincing and solid.
>
>
>
> A3: CACC Slowdown is a special case, because when a car crashes, every other car is assigned with a huge negative reward, which cannot be predicted with local observation. We suggest that this is probably the reason why reward prediction fails in CACC Slowdown.
>
>
>
> [1] Kaiser, Lukasz, et al. "Model-based reinforcement learning for atari." *arXiv preprint arXiv:1903.00374* (2019). https://arxiv.org/pdf/1903.00374.pdf
>
> [2] Wang, Tingwu, et al. "Benchmarking model-based reinforcement learning." *arXiv preprint arXiv:1907.02057* (2019). https://arxiv.org/abs/1907.02057.pdf

---

### Decision · Program_Chairs · 2022-01-20

**Decision:**

Reject

**Comment:**

As pointed out by reviewers, the presentation needs to be improved to clarify the algorithmic and theoretical contributions.